# Biochemical Characterization of an Endoglucanase GH7 from Thermophile *Thermothielavioides terrestris* Expressed on *Aspergillus nidulans*

Robson C. Alnoch [1,2,*], Jose C. S. Salgado [3], Gabriela S. Alves [2], Diandra de Andrades [1], Luana P. Meleiro [3], Fernando Segato [4], Gabriela Leila Berto [4], Richard J. Ward [3], Marcos S. Buckeridge [5] and Maria de Lourdes T. M. Polizeli [1,2,*]

[1] Departamento de Biologia, Faculdade de Filosofia, Ciências e Letras de Ribeirão Preto, Universidade de São Paulo, Ribeirão Preto 14040-901, SP, Brazil

[2] Departamento de Bioquímica e Imunologia, Faculdade de Medicina de Ribeirão Preto, Universidade de São Paulo, Ribeirão Preto 14049-900, SP, Brazil

[3] Departamento de Química, Faculdade de Filosofia, Ciências e Letras de Ribeirão Preto, Universidade de São Paulo, Ribeirão Preto 14040-901, SP, Brazil

[4] Departamento de Biotecnologia, Escola de Engenharia de Lorena, Universidade de São Paulo, Lorena 12602-810, SP, Brazil

[5] Laboratório de Fisiologia Ecológica de Plantas, Departamento de Botânica, Instituto de Biociências, Universidade de São Paulo, São Paulo 05508-090, SP, Brazil

[*] Correspondence: robsonalnoch@usp.br (R.C.A.); polizeli@ffclrp.usp.br (M.d.L.T.M.P.)

**Abstract:** Endoglucanases (EC 3.2.1.4) are important enzymes involved in the hydrolysis of cellulose, acting randomly in the β-1,4-glycosidic bonds present in the amorphous regions of the polysaccharide chain. These biocatalysts have been classified into 14 glycosyl hydrolase (GH) families. The GH7 family is of particular interest since it may act on a broad range of substrates, including cellulose, β-glucan, and xylan, an attractive feature for biotechnological applications, especially in the renewable energy field. In the current work, a gene from the thermophilic fungus *Thermothielavioides terrestris*, encoding an endoglucanase GH7 (*Tt*Cel7B), was cloned in the secretion vector pEXPYR and transformed into the high-protein-producing strain *Aspergillus nidulans* A773. Purified *Tt*Cel7B has a molecular weight of approximately 66 kDa, evidenced by SDS-PAGE. Circular dichroism confirmed the high β-strand content consistent with the canonical GH7 family β-jellyroll fold, also observed in the 3D homology model of *Tt*Cel7B. Biochemical characterization assays showed that *Tt*Cel7B was active over a wide range of pH values (3.5–7.0) and temperatures (45–70 °C), with the highest activity at pH 4.0 and 65 °C. *Tt*Cel7B also was stable over a wide range of pH values (3.5–9.0), maintaining more than 80% of its activity after 24 h. The $K_M$ and Vmax values in low-viscosity carboxymethylcellulose were 9.3 mg mL$^{-1}$ and $2.5 \times 10^4$ U mg$^{-1}$, respectively. The results obtained in this work provide a basis for the development of applications of recombinant *Tt*Cel7B in the renewable energy field.

**Keywords:** *Thermothielavioides terrestris*; endoglucanases; glycosyl hydrolase GH7; heterologous expression; *Aspergillus nidulans*; renewable energy

## 1. Introduction

Lignocellulosic biomass (LB) represents a potential replacement for the finite and highly polluting fossil fuels that are commonly used as energy sources. It may be obtained from plant matter such as wood chips and agricultural waste, which are abundant, widely available, inexpensive, and renewable [1–3]. This type of biomass can be converted into a variety of energy forms, including biofuels, electricity, and heat, making it a flexible and sustainable energy source. The LB components, such as cellulose, hemicellulose, and lignin, can be converted into multiple high-value products through processes in biorefineries. This combination of product generation and bioenergy co-production reinforces the concept of

an integrated lignocellulosic biorefinery approach, offering a more comprehensive solution for sustainable energy generation and resource utilization [4,5].

Cellulose is the most abundant natural polysaccharide on Earth and the essential structural component of the plant cell walls of lignocellulosic biomass [3,6]. It is a linear homopolymer of D-glucose units linked through β-1,4-glycosidic bonds [7]. The hydroxyl groups of the D-glucose units form strong networks of intra- and intermolecular hydrogen and Van der Waals bonds, resulting in a highly ordered structure known as the crystalline region. Moreover, the structure also presents disordered regions, arrayed irregularly, forming the amorphous regions [8,9].

Complete hydrolysis of cellulose requires the activity of several enzymes acting cooperatively to convert the homopolymer into fermentable sugars. Endo-1,4-β-glucanases (EC 3.2.1.4) are the main enzymes of the cellulolytic system, cleaving β-1,4 glycosidic bonds from cellulose. Endo-1,4-β-glucanases are responsible for releasing cello-oligosaccharides (COS) with different degrees of polymerization (DP), exhibiting a random degradation pattern, and creating new ends for the action of exo-β-1,4-glucanases (cellobiohydrolases or CBHs). CBHs (EC 3.2.1.91 and 3.2.1.176) hydrolyze the cellulose chains by the ends (reduced and non-reduced), releasing cellobiose as the main product. Finally, β-glucosidases (EC 3.2.1.21) convert COS and cellobiose into glucose [10–13]. The auxiliary activity (AA) enzymes such as lytic polysaccharide monooxygenases (LPMOs) (AA9, formerly GH61) are also important since these proteins can break the cellulose chain through an oxidative mechanism that is copper-dependent (EC 1.14.99.54 and EC 1.14.99.56), as well as carbohydrate-specific oxidoreductases such as the cellobiose dehydrogenase (CDH) (AA3) and cello-oligosaccharide dehydrogenase (AA7) (EC 1.1.99.18 and EC 1.1.99.-, respectively). AA3 and AA7 catalyze the oxidation of the reducing end C1-OH in cellobiose and COS to the corresponding lactones, donating electrons to LPMOs in the process [14].

The interest in endo-β-1,4-glucanases has grown remarkably due to their wide range of industrial applications, such as the food, feed, juice and beverage, textile, laundry, and pulp and paper industries. In addition, they have been heavily studied aiming at the enzymatic degradation of lignocellulosic materials in biorefineries to produce bioethanol [15–20]. Endoglucanases are found in 14 GH families: GH5 to GH10, GH12, GH26, GH44, GH45, GH48, GH51, GH74, and GH124. The endo-β-1,4-glucanases from the GH7 family are known for their promiscuous activity, acting on different substrates, including cellulose, β-glucan, lichenin, laminarin, and even xylan [21]. Therefore, there has been a growing attempt to identify and characterize novel endo-β-1,4-glucanases with improved properties for industrial use. It is necessary to overcome the limitations of current endo-β-1,4-glucanases and advance the field of industrial biotechnology through the development of more efficient and effective enzymes.

*Thermothielavioides terrestris* (formerly *Thielavia terrestris*) is a thermophilic fungus belonging to the class Sordariomycetes. It is known for its notable ability to grow in temperatures above 45 °C and produce extracellular enzymes, being classified as a good producer of thermostable enzymes of industrial interest, including cellulases and hemicellulases [22]. Among the CAZymes, different hydrolases GH7 (endo- and exo-glucanases) were identified and annotated depending on the strain of *T. terrestris* analyzed [23]. Nevertheless, the advancement of methods to enhance the yield and efficiency of fungal enzyme production could have substantial impacts on the economy and environment. Therefore, the heterologous protein expression by microbial systems is attractive due to their fast growth on inexpensive substrates, well-known genetics and physiology, and the availability of host strains [24]. Additionally, the use of filamentous fungi as hosts has benefits, such as their ability to secrete large amounts of protein and efficiently fold and modify it post-transduction [25]. Within this framework, in the current work, we report the cloning, production, purification, kinetics, and biochemical and structural characterization of an uncharacterized GH7 endoglucanase of *T. terrestris* (*Tt*Cel7B) expressed in heterologous host *Aspergillus nidulans* A773.

## 2. Results and Discussion

### 2.1. Sequence Analysis and Molecular 3D Modeling of TtCel7B

The protein sequence of *Tt*Cel7B was identified from CDS localized in *T. terrestris* chromosome III (AEO67421.1). *Tt*Cel7B has 464 amino acids, and domain analysis carried out using Pfam [26] showed the canonical catalytic domain (CCD) of the cellulase GH7 family (residues Gln[23] to Thr[397]) and a C-terminal cellulose-binding module from family 1 (CBM1) positioned between the residues Ala[428] and Leu[464], united by a 29-peptide linker (residues Val[398] to Thr[427]). A putative secretion signal sequence was annotated using the SignalP server [27] in the amino-terminal region, with a possible cleavage site between the residues 22 and 23. Structure-based amino acid sequence alignment of the CCD of *Tt*Cel7B with endoglucanases GH7 deposited in the Protein Data Bank (PDB) showed that putative CCD is composed of twenty-five β-strands, with four α-helices, aligned according to template structures (Figure 1).

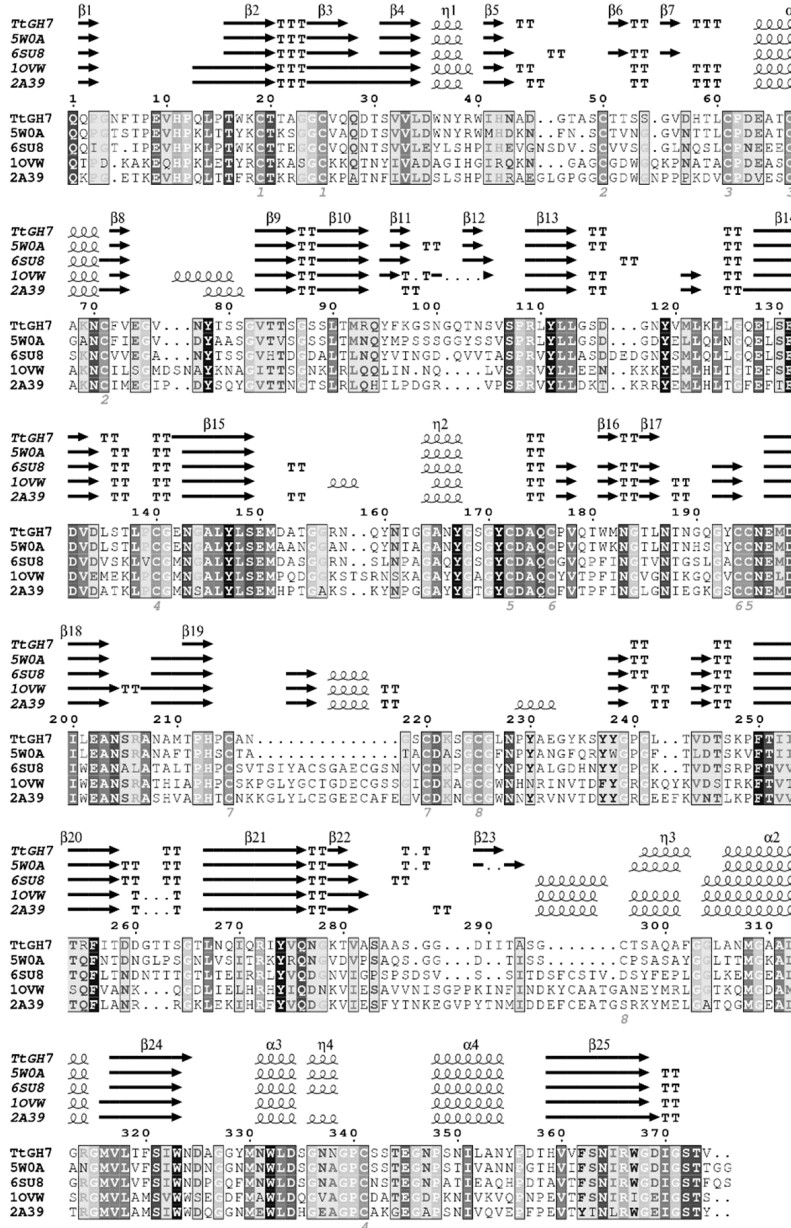

**Figure 1.** Multiple sequence alignment and secondary structure estimation of the endoglucanase *Tt*Cel7B (TtGH7) of *T. terrestris* and PDB models from *Trichoderma harzianum Cel7B* (PDB id 5W0A),

*Talaromyces emersonii* (PDB id 6SU8), *Fusarium oxysporum* (PDB id 1OVW), and *Humicola insolens* (PDB id 2A39). Dark- and light-shaded boxes indicate conserved residues. Secondary structure representations: α-helices are represented by α, arrows represent β-strands, and β-turns are marked with TT. The numbered positions indicate the disulfide bonds (Cys residues) predicted. Sequence alignments were performed using Clustal [28], and the figure was prepared with ESPriptS.

Conserved regions and residues were highlighted in boxes in the *Tt*Cel7B sequence, including the representative peptide sequence CNEMDIxEAN[195–204] (where X represents any amino acid), highly conserved in the endoglucanases GH7 (Figure 1) [29]. Among twenty cysteines residues identified in the *Tt*Cel7B sequence, twenty-six were predicted to form eight disulfide bridges (Cys[19–25], Cys[50–72], Cys[61–67], Cys[140–341] Cys[172–195], Cys[176–194] Cys[215–220], and Cys[225–296]), in analogous positions compared with *Cel7B* of *T. harzianum* [30], and other endoglucanases used as templates (Figure 1).

Phylogenetic analysis classified *Tt*Cel7B close to the endoglucanases of *Thermothelomyces thermophilus* (previously *Myceliophthora thermophila*), *Coniochaeta pulveracea*, and *Aspergillus fumigatus* (Figure 2A). The analysis of their sequences showed high identity (74.7, 74.2, and 74.2%, respectively) with these enzymes. Among the GH7 that have a known 3D structure, *Tt*Cel7B showed identity of 74, 73, and 59% with GH7 from *T. reesei* [PDB id 1GLM], *T. harzianum* [PDB id 5W0A], and *Rasamsonia emersonii* (*T. emersonii*) [PDB id 6SU8], respectively. The catalytic domain sequence of *Tt*Cel7B was structurally aligned with the Cel7B sequence [PDB id 5W0A], and the model was built without an amino acid sequence corresponding to the signal peptide, peptide linker, and CBM1. The final 3D model proposed with 374 residues was extensively checked and presents a ReFOLD global model quality score of 0.7010 (*p*-value < 0.001), z-DOPE score of −1.846, and GA341 score of 1.000, indicating a correct fold of structure [31–33].

The 3D model of *Tt*Cel7B was superimposed on the structures of the three GH7 used as templates (Figure 2B). The CCD displays a topology typical to the GH7 family, a β-jelly roll forming a distorted β-sandwich, similar to the structurally homologous templates, including shorter flexible loops covering the catalytic tunnel area (Figure 2B,C) [34–36]. The six most relevant loops had been annotated according to nomenclature A1, A2 A3, B1, B3, and B4 in the template structures (Figure 2B). The B1, A1, A2, and A3 loops of *Tt*Cel7B are properly aligned with other EG templates. These loops have an important role in the cellulose binding in the catalytic tunnel [37,38]. On the other hand, structural analysis suggests that the B3 and B4 loops are truncated in *Tt*Cel7B, as with Cel7B of *T. reesei* and *T. harzianum* [PDB id 5W0A] (Figures 1 and 2C) [30]. Putative *N*-glycosylation sites were annotated according to template ThCel7B [PDB id 5W0A], including the conserved site at Asn[183] (Figure 1), an important glycosylation site for the correct orientation of substrates and products in the catalytic tunnel of the enzyme [30]. Other critical residues to the coordination and orientation of long-chain substrates into the catalytic tunnel, such as Tyr[171], Tyr[147], Trp[323], and Trp[332] residues [29,35], were adequately aligned with the Cel7B template (Figure 2E). The conserved triad catalytic (Glu[202], Asp[199], and Glu[197]) was properly aligned with similar residues, with a distance within ~5.4 Å between side chains of Glu residues (Figure 2E), which suggests that *Tt*Cel7B performs the standard retaining catalytic mechanism of hydrolysis, well described for GH7 cellulases [30,38].

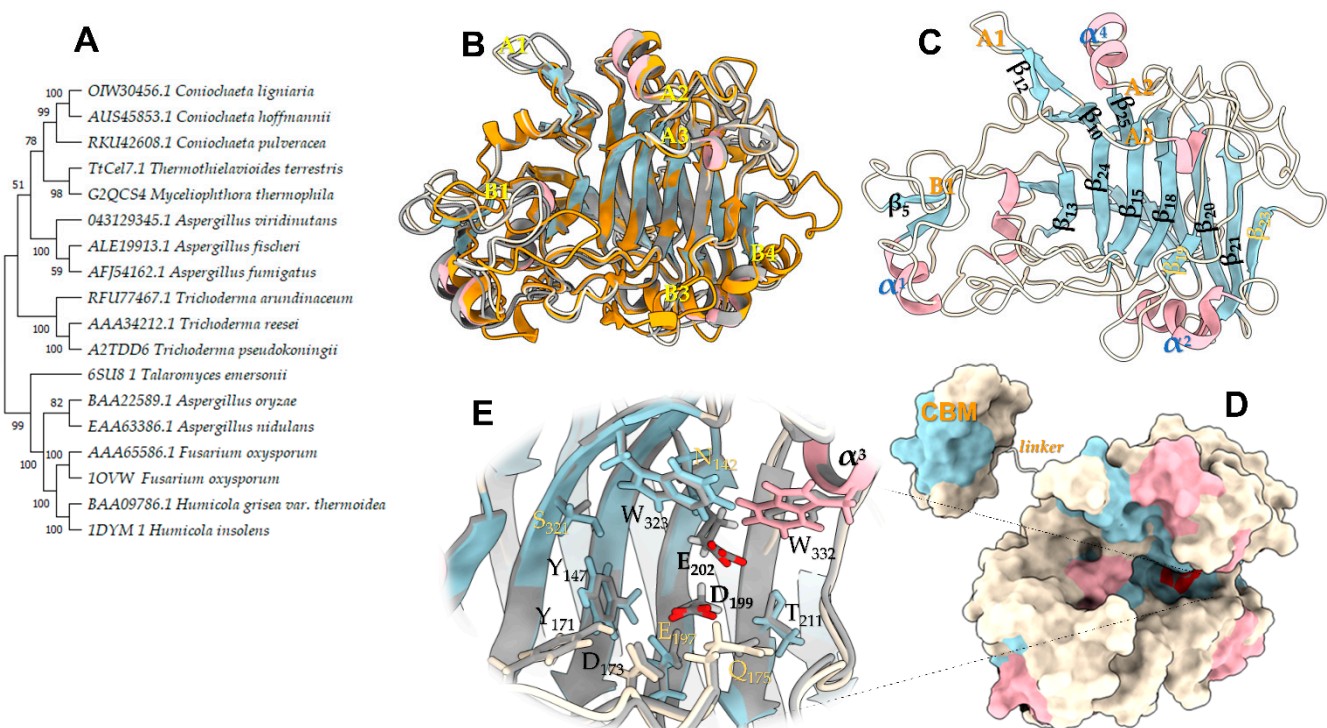

**Figure 2.** Phylogenetic analysis and homology model of the endoglucanase *Tt*Cel7B. (**A**) Phylogenetic tree of *Tt*Cel7B (*Tt*Cel7.1) and close CAZy GH7 proteins. The sequences were retrieved from GenBank, Uniprot, and PDB. (**B**) A 3D model of *Tt*Cel7B of superimposed with template structures from *T. harzianum* [PDB id 5W0A], *R. emersonii* [PDB id 6SU8], and *F. oxysporum* [PDB id 1OVW]. (**C**) Modeled structure, in the same orientation as in panel (**B**), with the β-strands, α-helices, and main loops highlighted. (**D**) Space-filling model of panel (**C**) linked with the cellulose-binding module (CBM). (**E**) Zoom-in of the modeled catalytic tunnel. Figure was prepared using UCSF ChimeraX.

## 2.2. Expression and Purification of Recombinant TtCel7B Endoglucanase

In the current work, we used the vector pEXPYR and the *A. nidulans* A773 strain as a heterologous host for the expression of *Tt*Cel7B. This expression system allows the overexpression along with the high-yield secretion and accumulation of the recombinant client protein with its native signal peptide in the culture supernatant [39]. Thus, the gene encoding endoglucanase *Tt*Cel7B was successfully cloned by Gibson Assembly into expression vector pEXPYR and transformed into *A. nidulans* A773 [39,40]. The resulting recombinant strains were selected by the reversion of the auxotrophic mark that allows growth in the absence of uracil and uridine, and the expression and secretion of *Tt*Cel7B in the culture supernatant were evaluated according to previous works [39]. *A. nidulans* A773 was selected and was cultivated in a minimal medium supplemented with pyridoxine and 5% of maltose for 48 h at 37 °C. The culture supernatant containing *Tt*Cel7B was concentrated and purified in two chromatography steps using a Hiprep Q 16/10 Fast Flow column (Figure 3A), followed by size exclusion in a Superdex 75 16/60 column. *Tt*Cel7B was successfully purified, as evidenced by a single band in SDS-PAGE, with a molecular weight (MW) estimated at 69.5 kDa (Figure 3B). Its MW value is higher than the theoretical value of 48.7 kDa predicted in silico by the ProtParam software-based amino acid sequence. Thus, mass spectrometry analysis was carried out to confirm the identity of *Tt*Cel7B. The fragment mass data obtained were analyzed and compared to theoretical masses predicted by the in silico peptide cleavage of the *Tt*Cel7B sequence (Supplementary Table S1), confirming the identity of *Tt*Cel7B (AEO67421.1). Therefore, these results indicated that the MW estimated of 69.5 kDa for *Tt*Cel7B corresponds to the glycosylated form. Protein glycosylation is usually observed in endoglucanases expressed by filamentous fungus [41], as recently reported for GH7 ThCel7B (*T. harzianum*) expressed

in *Aspergillus niger* PY11 [42], Af-EGL7 (*A. fumigatus*) expressed in *P. pastoris* X-33 [43], ReCel7B (*R. emersonii*) expressed in *Aspergillus oryzae* [35], and for MtGH45 (*M. thermophila*) expressed in *A. nidulans* A773 [44].

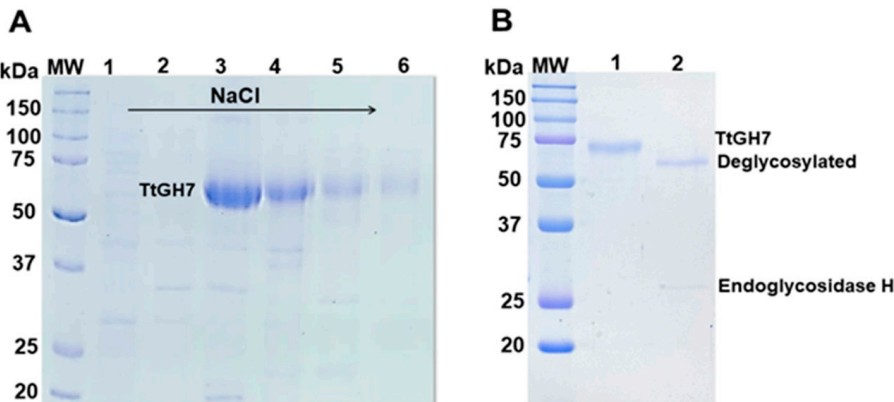

**Figure 3.** Purification of the recombinant endoglucanase *Tt*Cel7B from *T. terrestris* expressed in *A. nidulans* A773. (**A**) SDS-PAGE fractions (lines 1–6) eluted from the ion-exchange Hiprep Q-FF chromatography column. (**B**) SDS-PAGE of *Tt*Cel7B purified (line 1) from size exclusion Superdex 75 10/300 GL column and analysis of the enzymatic deglycosylation assay (lane 2), showing the purified *Tt*Cel7B after digestion treatment with endoglycosidase H. MW: molecular weight standard ladder is shown as kDa.

*2.3. Biochemical Characterization*

2.3.1. Estimation of Carbohydrate Content of *Tt*Cel7B

To estimate the carbohydrate content of purified *Tt*Cel7B, deglycosylation assays (*N*-glycosylation) were performed using the endoglycosidase H from *Streptomyces plicatus*. SDS-PAGE shows the profile of native and deglycosylated *Tt*Cel7B, with a MW estimated of 65.2 kDa, resulting in carbohydrate content of approximately 6% (Figure 3B). These results corroborate the in silico analysis performed by the NetGlyc 1.0 Server and the *N*-glycosylation sites predicted from the 3D model of *Tt*Cel7B (Figure 2). On the other hand, the MW of deglycosylated *Tt*Cel7B is still higher than the theoretical value (48.7 kDa), which suggests a high level of *O*-glycosylation in *Tt*Cel7B, probably along the flexible peptide linker and CBM of the enzyme, a region rich in Thr and Ser residues (Figure 1), potential *O*-glycosylation sites [43,45–47]. Therefore, the high yield of secretion and higher molecular weight of recombinant *Tt*Cel7B show that the expression system of *A. nidulans* A773 promoted correct post-translational modifications on the enzyme.

2.3.2. Circular Dichroism (CD) and Determination of Melting Temperature (Tm) of *Tt*Cel7B

Circular dichroism spectroscopy analysis of *Tt*Cel7B showed a typical β-class protein profile spectrum, with the minimum negative band at 214 nm and the absence of a negative band at 222 nm (Figure 4A), suggesting low content of α-helices [42,48–50]. Analyses of the CD spectral data with the DichroWeb server [51] showed regular content of β-strands and α-helices of 53% and 8%, respectively. In addition, the Differential Scanning Fluorimetry (DSF) technique was used for the determination of the melting temperature ($T_m$) of *Tt*Cel7B at different pH levels. As shown in Figure 4B, there was a variation in the $T_m$ depending on the pH, with minimal values at pH 3.0 ($T_m$ 48.9 $\pm$ 0.1) and maximum at pH 8.0 ($T_m$ 66.3 $\pm$ 0.4), indicating a strong effect of pH on the *Tt*Cel7B structure (Figure 4B). Similar results for the effect of pH and temperature on the secondary structure were reported for GH7 Cel7A and Cel7B of *T. harzianum* [42,50].

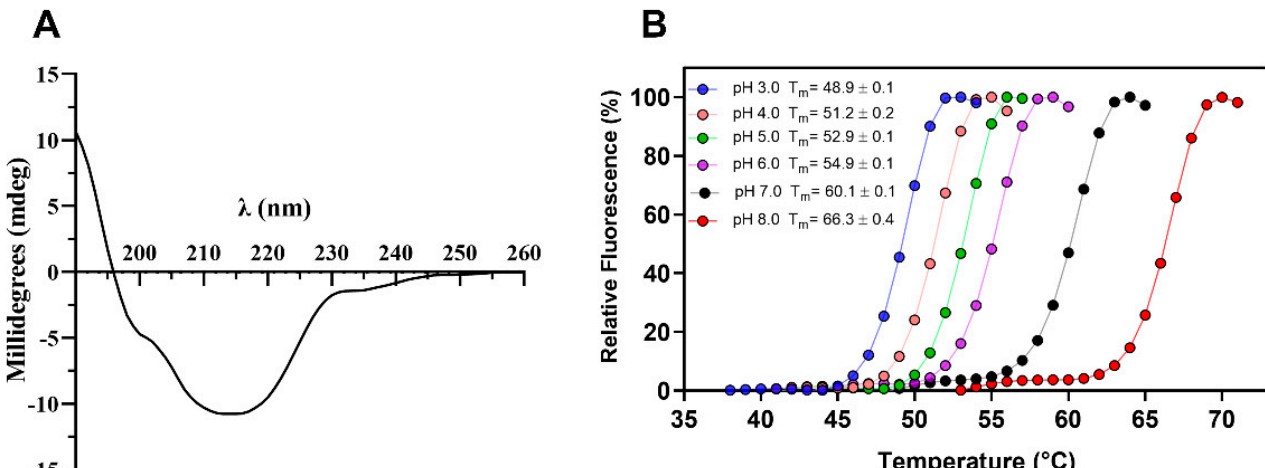

**Figure 4.** (**A**) Secondary structure profile of *Tt*Cel7B determined by CD spectroscopy. (**B**) Thermal stability curves of *Tt*Cel7B at different pH and determination of melting temperature (Tm).

### 2.3.3. Panel Substrates and Effect of Different Compounds on *Tt*Cel7B Activity

Endoglucanases GH7 are known for their promiscuous activity, acting on different substrates in addition to the cellulose, such as β-glucan, lichenin, xylan, and xyloglucan [21]. To characterize the substrate specificity of *Tt*Cel7B, we determined the activity of the enzyme against several polysaccharide substrates. As shown in Figure 5A, *Tt*Cel7B hydrolyzed different substrates from CMC to galactomannan (locust bean gum). Although the highest activity was against CMC of low viscosity, $331 \pm 5$ U mg$^{-1}$, it also had high activity against xyloglucan from tamarind ($294 \pm 20$ U mg$^{-1}$) and β-glucan from barley ($211 \pm 4$ U mg$^{-1}$) (Figure 5A). It is not surprising that *Tt*Cel7B had significantly different activities against medium- and low-viscosity CMC, since differences in the degree of polymerization (DP) and degree of substitution (DS) are observed in the medium- and low-viscosity CMC [52,53]. For example, endoglucanase MtEG7 from *M. thermophila* was shown to be more active in barley β-glucan than CMC, and the possible reason is that CMC is substituted with methoxy side chains, and this could interfere with the enzyme activity against the CMC [54]. *Tt*Cel7B showed low activity against locust bean gum, and did not display activity in xylan or arabinan, which indicated higher specificity for polysaccharides β-1,4 and β-1,3-linked substrates containing glucan monomers, such as cellulose, xyloglucan, and β-glucan. Recently, one cellulase from *T. terrestris* Co3Bag1 was characterized (*Tt*Cel7A), showing excellent bifunctional cellulase/xylanase activity against CMC and beechwood xylan, but it also hydrolyzed oat bran, wheat bran, and sugarcane bagasse, releasing glucose and cellobiose as the main products [23]. *Tt*Cel7B also displayed activity against a broad range of polysaccharides, an important feature in the designed cocktail enzymatic, present in some GH7 of commercial interest. For example, the substrate specificity of *Tt*Cel7B is similar to that of Cel7B of *T. reesei* [21] and *T. harzianum* [42], and MtEG7a from *M. thermophila* [53].

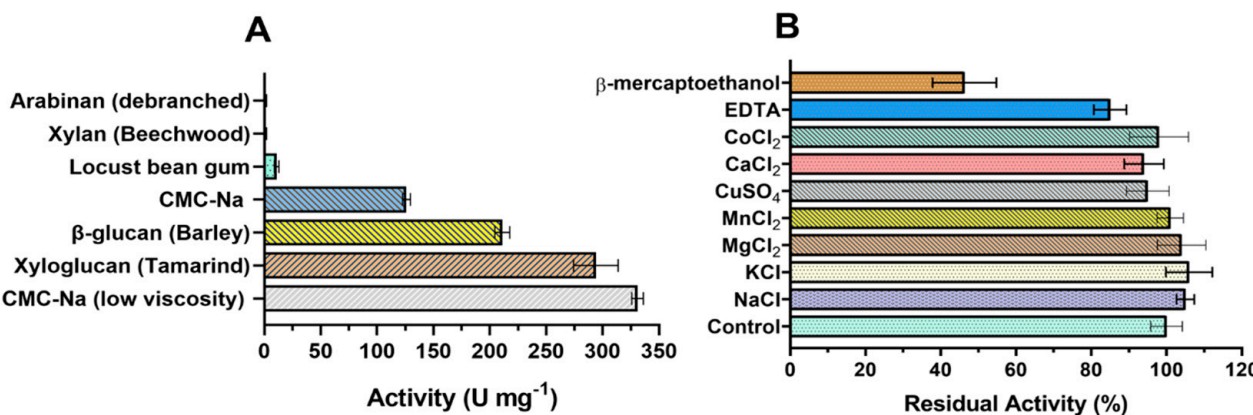

**Figure 5.** (**A**) Substrate specificity of the endoglucanase GH7 from *T. terrestris* (*Tt*Cel7B). (**B**) Effect of different ions and compounds on *Tt*Cel7B activity against medium-viscosity CMC-Na.

The effect of different ions on *Tt*Cel7B activity was evaluated at a final concentration of 5 mmol L$^{-1}$. As observed in Figure 5B, *Tt*Cel7B showed more than 90% relative activity for all ions evaluated. Interestingly, the negative effect of EDTA on *Tt*Cel7B activity was mild, retaining more than 80% of its relative activity against medium-viscosity CMC-Na (Figure 5B). On the other hand, a negative effect was observed on *Tt*Cel7B activity after the addition of β-mercaptoethanol, with a decrease >50% in the relative activity (Figure 5B). These results suggest a critical role of the disulfide bonds in the active structural conformation of *Tt*Cel7B. As described above, structural analysis of the 3D model of *Tt*Cel7B indicated eight putative disulfide bridges in the *Tt*Cel7B structure (Figures 1 and 2).

### 2.3.4. Effect of Temperature and pH on *Tt*Cel7B Activity and Stability

The effect of temperature on *Tt*Cel7B activity was determined by the hydrolysis of CMC over the range of 45 to 80 °C. The maximum endoglucanase activity was obtained at 65 °C, decreasing by 30% at 75 °C and more than 70% at 80 °C (Figure 6A). In the thermal stability studies, *Tt*Cel7B was most stable at 45 and 50 °C, with residual activities above 90% after 24 h of incubation. At the highest temperature, *Tt*Cel7B showed residual activity above 70 and 50% after 6 h of incubation at 60 and 65 °C, respectively (Figure 6C). In relation to the pH effect, *Tt*Cel7B was active over a wide range of pH values from 3.0 to 9.0, with maximum activity at pH 4.5 (Figure 6B). However, the activity decreases by more than 50% at a pH above 5.5, indicating a strong effect of pH on *Tt*Cel7B activity (Figure 6D). Interestingly, *Tt*Cel7B showed residual activities above 60% after being incubated at pH 3.0 to 9.0 for 24 h (Figure 6D). Thus, the activity at high-temperature and low-pH conditions confirmed that *Tt*Cel7B corresponds to an acidic thermophilic endoglucanase, with similar characteristics to those reported for MtEG7 of *M. thermophila* [53] and Cel7B of *T. harzianum* [42]. Cel7B (expressed on *A. niger*) showed maximum activity at 55 °C and pH 3.0, and, surprisingly, retained 100% of its residual activity after two months of incubation at 55 °C [42]. MtEG7 (expressed in *Pichia pastoris*) showed maximum activity at 60 °C and pH 5.0, with residual activities around 50% after 6 h of incubation at 70 and 80 °C [53].

Other recombinant endoglucanases with similar properties expressed in *P. pastoris* have been reported (Table 1). For example, Af-EGL7 from *Aspergillus fumigatus* showed maximum activity at 55 °C, at pH 5.0, and retained 100% of its residual activity after 72 h of incubation at 55 °C [43]. For Cel7A from *Neosartorya fischerii*, the maximum activity was reported at 60 °C, pH 5.0 [55]. Egl7A from *T. emersonii* presented maximum activity at 70 °C, at a pH of 4.5, and high stability over a range of pH values (1.0–11), with residual activities >70% after 1 h of incubation [56]. CTendo7 from *Chaetomium thermophilum* showed the best enzyme activity at 55 °C and pH 5.0, showing 61.3% of activity after 60 min at 80 °C, with almost all activity lost after 100 min at 90 °C [47].

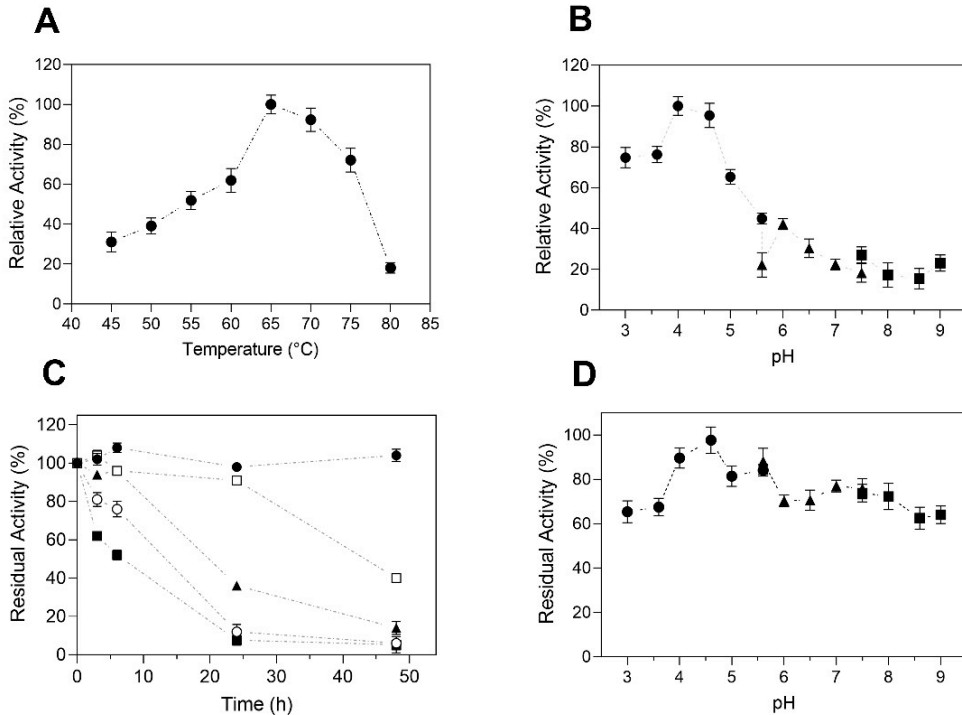

**Figure 6.** Effect of temperature and pH on *Tt*Cel7B activity and stability. (**A**) The effect of temperature on activity was determined with CMC as the substrate, at pH 4.0; (**B**) effect of pH on activity, at 65 °C. Key: (●) citrate buffer, (▲) phosphate buffer, (■) tris-HCl buffer. (**C**) *Tt*Cel7B was incubated for 48 h at temperatures of 45 °C (●), 50 °C (□), 55 °C (▲), 60 °C (○), and 65 °C (■). Residual activity corresponding to the activity before incubation (100%); (**D**) stability at different pH values for 24 h, with 100% of residual activity corresponding to the activity at pH 4.0. Key: (●) citrate buffer, (▲) phosphate buffer, (■) tris-HCl buffer. Results were expressed as the average of triplicate assays ± the standard error of the mean.

**Table 1.** General catalytic properties of *Tt*Cel7B compared with other endoglucanases.

| Source | Expression System | Substrate | Temperature/ pH Optimum | Vmax (U mg$^{-1}$) | $K_M$ (mg mL$^{-1}$) | Kcat (s$^{-1}$) | Kcat/$K_M$ (mL mg$^{-1}$ s$^{-1}$) |
|---|---|---|---|---|---|---|---|
| *T. terrestris* (This work) | *A. nidulans* | CMC-Na | 65 °C/4.5 | 327.2 ± 16.4 | 9.49 ± 0.48 | 358.28 ± 17.9 | 37.74 ±1.9 |
| *A. fumigatus* [43] | *P. pastoris* | CMC-Na | 55 °C/5 | 6193 ± 140 | 24.5 ± 0.6 | 5037 ± 114 | 205.9 ± 0.8 |
| *A. fumigatus* [57] | *Escherichia coli* | CMC-Na | 55 °C/5 | 51.9 ± 0.007 | 209.7 ± 0.1 | 43.3 | 0.2 |
| *C. thermophilum* [47] | *P. pastoris* | CMC-Na | 55 °C/5 | 59.6 ± 8.2 | 79.2 ± 5.8 | $2.11 \times 10^{-3}$ | 0.02673 |
| *M. thermophila* [53] | *P. pastoris* | CMC-Na | 60 °C/5 | 622.5 ± 86.4 | 24 ± 0.5 | - | 18.82 |
| *T. harzianum* [42] | *A. niger* | Xyloglucan | 55 °C/3 | 0.22 * | 1.98 ± 0.47 | 0.45 | - |

\* The Vmax for this substrate was expressed as $\mu M \cdot s^{-1}$.

## 2.4. Kinetic Parameters

The *Tt*Cel7B presented Vmax = 327.2 ± 16.4 U mg$^{-1}$, $K_M$ = 9.49 ± 0.48 mg ml$^{-1}$, kcat = 358.28 ± 17.9 s$^{-1}$, and kcat/$K_M$ = 37.74 ±1.9 mL mg$^{-1}$ s$^{-1}$ using low-viscosity CMC-Na as a substrate (Table 1). Other heterologous GH7 produced in different expression systems have a wide range of kinetic parameter values (Table 1). *C. thermophilum* (CTendo7) has Vmax and $K_M$ of 59.6 ± 8.2 U mg$^{-1}$ and 79.2 ± 5.8 mg ml$^{-1}$, respectively; on the other hand, MtEG7a from *M. thermophila* has Vmax of 622.5 ± 86.4 U mg$^{-1}$ and of $K_M$ 4.5 mg ml$^{-1}$ on CMC-Na [47,53]. The GH7 from *Aspergillus fumigatus* (Af-EGL7) expressed

in *P. pastoris* has Vmax and $K_M$ of 6193 $\pm$ 140 U mg$^{-1}$ and 24.5 $\pm$ 0.6 mg ml$^{-1}$, respectively, on CMC-Na [43]. On the other hand, the same enzyme expressed on *E. coli* presented Vmax and $K_M$ of 51.9 $\pm$ 0.007 U mg$^{-1}$ and 209.7 $\pm$ 0.1 mg ml$^{-1}$, respectively [57]. This shows that the appropriate expression system could be crucial when aiming at industrial applications, since the same enzyme could have very different catalytic efficiency (kcat/$K_M$) when expressed in different systems (for example, 205.9 $\pm$ 0.8 for *P. pastoris* and 0.2 mL mg$^{-1}$ s$^{-1}$ for *E. coli* to Af-EGL7). *T. harzianum* endoglucanase I GH7 (Cel7B) has kinetic parameters measured on xyloglucan (Vmax of 0.22 $\mu$M s$^{-1}$ and $K_M$ 1.98 $\pm$ 0.47 mg ml$^{-1}$), despite being very well known as an endoglucanase that acts on CMC-Na [42]. Enzymes from the GH7 family can act in different substrates, such as cellulose, $\beta$-glucan, lichenin, laminarin, and even xylan, showing promiscuous activity on substrates [42,43,47,53,55]. This characteristic could be interesting for the industrial application of *Tt*Cel7B, since this enzyme could be used to produce different products from distinct sources.

## 3. Materials and Methods

### 3.1. Strains, Reagents, and Materials

The *A. nidulans* strain A773 (*pyrG89*; *wA3*; *pyroA4*) was obtained from the Fungal Genetic Stock Center (San Luis, MO, USA) and *T. terrestris* UAMH 3264 was purchased at the Microfungus Collection of the University of Alberta (Edmonton, AB, Canada).

A Miracloth membrane, 3,5-dinitrosalicilic acid (DNS), carboxymethylcellulose (CMC), and glucose (>95%), were purchased from Sigma-Aldrich (St. Louis, MO, USA). Locust bean, debranched arabinan, $\beta$-glucan, and xyloglucan were acquired from Megazyme (Bray, Wicklow, Ireland). Bradford protein assay and Precision Plus protein T.M. standards were purchased from Bio-Rad Laboratories (Hercules, CA, USA). Superdex 75 10/300 GL and Hiprep Q FF Superdex 75 10/300 GL columns were acquired from G.E. Healthcare (Uppsala, Sweden). All other reagents used for the assays were of analytical grade.

### 3.2. Sequence Analysis and Phylogenetic and Structural Homology Modeling

Multiple protein sequence alignment was carried out with the ClustalW algorithm [28] using sequences obtained from NCBI (National Center for Biotechnology Information) [58], Uniprot (Universal Protein Resource) [59], and PDB (Protein Data Bank) [60]. Selected PDB templates were used to predict the secondary structure and features of the amino acid sequence using the ENDscript server [61]. A phylogenetic tree was built with the neighbor-joining method using the MEGA software (version 11), with bootstrapping of 1000 replicates [62]. The 3D model of *Tt*Cel7B was built by homology using the MODELLER software [63]. The templates *T. reesei* [PDB id 1GLM] and *T. harzianum* [PDB id 5W0A] were acquired from PDB. Five hundred models were generated and optimized by the VTFM method within MODELLER and one was considered the final model according to the embedded normalized DOPE [64] and GA3415 scores [65]. The quality of the final model was assessed by the programs ReFOLD [32,33] and PROCHECK [66]. The AlphaFold database [67] embedded in UCSF ChimeraX [68] was used to predict the *Tt*Cel7B 3D model with a peptide linker and carbohydrate-binding module (CBM) structure.

### 3.3. Cloning, Expression, and Purification of the Endoglucanase TtCel7B

Extraction of the genomic DNA of *T. terrestris* was carried out using the kit Wizard Genomic DNA Purification (Promega, Madison, WI, USA), in accordance with the instructions of the manufacturer. Primers were designed based on the coding sequence of *Tt*Cel7B and amplified from genomic DNA by polymerase chain reaction (PCR) with the oligonucleotides TtGH7 fwd **CATTACACCTCAGCA**ATGGGCCAGAAGACGCTGCACGGATT and TtGH7 rev **GTCCCGTGCCGGTTA**TTAGAGGCACTGGTAGTACCAGGGGTTCAG (Exxtend Oligo Solutions) using Phusion High-Fidelity DNA Polymerase (New England Biolabs, Ipswich, MA, USA). The amplified genes were used to assemble the pEXPYR vector via the Gibson Assembly Method using the complementary regions (in bold) [69], and the recombinant pEXPYR-*Tt*Cel7B construct was used to transform *A. nidulans* A773,

as previously described [39]. The *A. nidulans* recombinant strains were further selected after growing in the absence of uracil and uridine in the minimal medium, and the expression of *Tt*Cel7B was confirmed by SDS-PAGE [39].

Heterologous expression of *Tt*Cel7B was carried out according to Damásio et al. (2012) [70] and Segato et al. (2017) [71]. Briefly, a spore suspension ($10^8$ spores mL$^{-1}$) of recombinant strain was inoculated in 500 mL of minimal medium supplemented with 5% maltose (*w/v*), pH 6.5, and incubated at 37 °C for 48 h. Afterward, the crude extract was filtered in a Büchner funnel with a Miracloth$^®$ membrane, followed by concentration and dialysis in 50 mmol L$^{-1}$ phosphate buffer, pH 6.0, using the QuixStand Benchtop Systems tangential concentrator with a 30 kDa Hollow Fiber Cartridge (GE Healthcare, Chicago, IL, USA). Next, the crude extract was loaded on an anion exchange column (Hiprep Q FF 16/10) equilibrated with 50 mmol L$^{-1}$ phosphate buffer, pH 6.0, integrated with an ÄKTA Purifier 900 (GE Healthcare) chromatography system. Protein elution was monitored at 280 nm, with a linear gradient from 0 to 1 mol L$^{-1}$ of NaCl. The protein samples of interest were selected by SDS-PAGE, pooled, and concentrated (30 kDa cutoff—Vivaspin GE Healthcare). The recovered sample was loaded in a size exclusion column (Superdex 75 10/300 GL), equilibrated with phosphate buffer 50 mmol L$^{-1}$, NaCl 150 mmol L$^{-1}$, pH 6.0. Protein elution was monitored at 280 nm, and the fractions of interest were selected as described above, and stored at −20 °C.

*3.4. Determination of Endoglucanase Activity*

Endoglucanase activity was determined by the reducing sugars method using 3,5 dinitrosalicylic acid (DNS) [72]. The assays were carried out with 50 μL of the enzyme, 50 μL of citrate buffer (0.1 mol L$^{-1}$, pH 5.0), and 100 μL of 1% CMC (*w/v*). The assay was performed for 10 min at 50 °C and 1300 rpm. After incubation, 200 μL of DNS was added to the assay mixture and boiled at 98 °C for 5 min. Afterward, samples of 100 μL were collected for the quantification of reducing sugars at 540 nm using microplate readers (Spectramax M2, Molecular Devices, San Jose, CA, USA). A standard curve of D-glucose was used to estimate the reducing sugar concentration equivalents. One unit of enzyme activity (U) was defined as the amount of enzyme that catalyzed the release of 1 μmoL of reducing sugar per minute under the assay conditions. Assay controls (blank) were performed without *Tt*Cel7B, which was added after the addition of DNS (prior boiling).

*3.5. Protein Content, Electrophoresis, and Deglycosylation Analysis*

Protein content was determined by the method of Bradford [73], using the Bradford Protein Assay Kit (Bio-Rad) with bovine serum albumin as the standard. Electrophoresis of protein samples (15% SDS-PAGE) was done according to Laemmli [74]. Gels were stained with Coomassie Brilliant Blue R and de-stained with a solution of ethanol/acetic acid/water (5/1/4 *v/v/v*). Glycosylation of *Tt*Cel7B was analyzed with Endoglycosidase H (EndoH) (Roche, Mannheim, Germany), using the manufacturer's instructions. Briefly, 5 μg of purified *Tt*Cel7B was mixed with 250 mU of EndoH in sodium acetate buffer 50 mM pH 5.5 and incubated at 37 °C overnight. The samples were analyzed by SDS-PAGE after treatment and the estimation of the carbohydrate content was performed from the migration difference between the treated and untreated *Tt*Cel7B compared to the molecular mass standard.

*3.6. Liquid Chromatography–Tandem Mass Spectrometry (LC–MS/MS)*

The spot of the purified *Tt*Cel7B from the Coomassie-stained SDS-PAGE was excised, reduced, alkylated, and submitted to in situ trypsin gel digestion (Promega). One microliter of tryptic peptides was applied in a C18 column (75 μm × 100 mm) for desalting. The peptides were analyzed by RP-nanoUPLC (nanoAcquity, Waters, Milford, MA, USA) coupled with a Q-TOF Ultimamass Spectrometer (Waters) with a nano-electrospray source at a flow rate of 0.6 μL min$^{-1}$. The gradient was set to 2–90% (*v/v*) acetonitrile in 0.1% (*v/v*) formic acid over 45 min. The equipment was set up on the top three modes. The software

MassLynx v.4.1 (Waters) was used to obtain the spectra, and the software Mascot Distiller v.2.3.2.0, 2009 (Matrix Science Ltd., London, UK) was used to convert the raw data files to a peak list format (mgf). The engine Mascot v.2.3 (Matrix Science Ltd.) was used to compare the MS/MS profiles against predicted protein sequences of *T. terrestris* [75,76].

### 3.7. Differential Scanning Fluorimetry (DSF) and Circular Dichroism (CD)

Differential Scanning Fluorimetry (DSF) was used for the determination of the melting temperature (Tm) of *Tt*Cel7B. In a 96-well PCR plate was added 2 µg of purified *Tt*Cel7B, 2 µL of SYPRO Orange solution (50x), and ultrapure water up to 20 µL in each well. The plate was sealed with Sealing Tape for Optical Assays (Bio-Rad #2239444) and centrifuged at $4000\times g$ for 2 min and 4 °C. The plates were heated from 20 to 90 °C in the thermocycler (in increments of 1 °C) with a scanning rate of 1 °C min$^{-1}$. The thermocycler lid was heated to 100 °C to avoid condensation effects during the experiment. The changes in the fluorescence in the plate wells were monitored simultaneously at 470 and 570 nm, which correspond to the excitation and emission lengths of SYPRO® Orange, respectively. The melting temperature ($T_m$) was calculated by nonlinear regression using a sigmoidal curve through the Boltzmann equation to adjust fluorescence data.

The analysis of the secondary structure of the proteins was performed by circular dichroism (CD) using a Jasco-810 spectropolarimeter (JASCO Inc., Tokyo, Japan). Protein samples (0.1 mg mL$^{-1}$) mixed in Tris-HCl buffer (10 mM) were added in a quartz cuvette of 200 µL, with an optical path length of 0.1 mm. The data were collected using a scanning speed of 50 nm/min, spectral bandwidth of 3 nm, and response time of 1 s. Buffer spectra without protein were subtracted in all experiments (blank), with all measurements performed with an average of six accumulations collected, in the range of far UV (190–250 nm).

### 3.8. Effect of Different Ions and Compounds on TtCel7B Activity

The effect of different ions on *Tt*Cel7B activity was evaluated at a final concentration of 5 mmol L$^{-1}$ in a standard activity assay. The ions added in salt form were $CoCl_2 \cdot 6H_2O$, $CaCl_2$, $CuSO_4$, $MnCl_2 \cdot 4H_2O$, $MgCl_2 \cdot 6H_2O$, KCl, and NaCl. Reducing agent β-mercaptoethanol and chelating agent EDTA (ethylenediaminetetraacetic acid) also were evaluated at the same final concentration. The relative activities were calculated concerning a control performed under a standard condition of the assay and attributed a relative activity of 100%. Results were expressed as the average of triplicate assays ± the standard error of the mean.

### 3.9. Effect of Temperature and pH on TtCel7B Activity and Stability

The effect of temperature on *Tt*Cel7B activity was determined by the hydrolysis of CMC-Na over the range of 45 to 80 °C, at pH 4.5. The thermal stability was assessed via the incubation of *Tt*Cel7B in a water bath for 48 h at temperatures ranging from 45 to 65 °C. The effect of pH on activity was analyzed over the range of 3.0 to 9.0 at 50 °C using 0.05 mol L$^{-1}$ of the buffers citrate (3.0 to 5.5), phosphate (5.5 to 7.5), and Tris-HCl (7.5 to 9.0). The pH stability of *Tt*Cel7B was determined by incubation for 24 h, at 25 °C, at different pH values, with the same buffers used above (3.0 to 9.0). The residual activities were calculated relative to the control, which was treated identically but without incubation and attributed a residual activity of 100%. Results were expressed as the average of triplicate assays ± the standard error of the mean.

### 3.10. Determination of Kinetic Parameters

The kinetic parameters of maximum velocity (Vmax) and Michaelis–Menten constant ($K_M$) were determined using low-viscosity carboxymethylcellulose (CMC-Na) (Sigma-Aldrich #C5678). The assay was carried out for 10 min in citrate buffer (50 mM) at pH 4.5, 65 °C, and CMC concentrations ranging from 0.02 to 32 mg ml$^{-1}$. The kinetic parameters were calculated by the software SigrafW [77] from the experimental data using

nonlinear regression (Hill equation). The values are presented here as the mathematical mean of three independent measurements from two protein preparations ($n = 6$).

## 4. Conclusions

In conclusion, the present work describes the cloning and characterization of an endoglucanase GH7 (*Tt*Cel7B) from the thermophilic fungus *Thermothielavioides terrestris.* The gene was successfully transformed into the high-protein-producing strain *Aspergillus nidulans* A773, and the purified enzyme showed a molecular weight of 66 kDa. Biochemical assays demonstrated that *Tt*Cel7B has a wide range of activity in terms of pH and temperature, with the highest activity at pH 4.0 and 65 °C, and it was also stable over a wide range of pH. Circular dichroism showed high content of β-strands, consistent with the canonical GH7 family β-jellyroll fold found in other GH7. The enzyme showed higher specificity for polysaccharides β-1,4 and β-1,3-linked substrates, such as cellulose, xyloglucan, and β-glucan. The promiscuous activity could be beneficial when aiming at the production of cocktails, since *Tt*Cel7B could act in different polysaccharides in lignocellulosic biomass hydrolysis. Finally, the results of this study provide a foundation for the further development of *Tt*Cel7B for applications in the renewable energy field.

**Supplementary Materials:** The following supporting information can be downloaded at: https://www.mdpi.com/article/10.3390/catal13030582/s1, Table S1. Amino acid sequence of the endoglucanase (*Tt*Cel7B) from *T. terrestris*. The peptides corresponding to those identified by LC/MS analyses are shown in bold. The mass profiles obtained were used in database MASCOT (Matrix Science, London, UK).

**Author Contributions:** Conceptualization, methodology, writing—original draft preparation: R.C.A., J.C.S.S., G.S.A. and D.d.A.; software, validation, formal analysis, investigation, visualization: R.C.A., J.C.S.S., G.S.A., D.d.A., L.P.M., R.J.W., M.S.B., G.L.B. and F.S.; data curation: R.C.A., G.S.A., J.C.S.S., D.d.A. and L.P.M.; review, resources and funding acquisition: R.C.A., R.J.W., M.S.B. and F.S.; writing—review and editing, resources, supervision, project administration, funding acquisition: M.d.L.T.M.P. All authors have read and agreed to the published version of the manuscript.

**Funding:** The authors gratefully acknowledge FAPESP (São Paulo Research Foundation, Grant No. 2018/07522-6, 2019/22284-7, 2021/06679-1, and 2022/042279) and FCT (POCI-01-0145-FEDER-032206)—transnational cooperation project EcoTech, and the National Institute of Science and Technology of Bioethanol, INCT, CNPq 465319/2014-9/FAPESP n° 2014/50884-5) for the financial support. Research scholarships were granted to R.C.A., D.A., L.P.M., and J.C.S.S. by FAPESP (Grant No. 2020/00081-4, 2020/15510-8, 2016/17582-0, and 2019/21989-7, respectively), CNPq 310340/2021-7 to M.L.T.M. Polizeli, and to G.S.A. by CAPES (Coordenação de Aperfeiçoamento de Pessoal de Nível Superior, Finance Code 001).

**Data Availability Statement:** Not applicable.

**Acknowledgments:** We acknowledge Mariana Cereira and Mauricio de Oliveira for the technical support.

**Conflicts of Interest:** The authors declare no conflict of interest.

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
