# Peer review of "Biochemical Characterization of an Endoglucanase GH7 from Thermophile Thermothielavioides terrestris Expressed on Aspergillus nidulans"

_catalysts, doi:10.3390/catal13030582_

Round 1

Reviewer 1 Report

This manuscript reports the recombinant expression in the Aspergillus nidulans platform of a GH7 endoglucanase from the fungus Therthielavioides terrestris and its biochemical characterization. The manuscript is straightforward and it is very well written.

However, there are some issues that need to be addressed before publishing.

Most relevant issues:

1-   Why was this particular sequence selected? Is it the only GH7 endoglucanase of the fungus? Please include some information in the introduction.

2-   Regarding the expression (section 2.2): was it expressed with its native signal peptide? Or the mature version, with the signal peptide of the vector?

3-   It is important to deposit or include the complete sequence of the recombinant protein. This will help to verify the results shown in Table 1 (mas spec peptides).

4-   Methods 3.3: Why was DNA used? Wasn´t the sequence amplified from RNA? What about the introns? It does not make sense. Please revise

5-   The last sentence of page 9 is a repetition of the introduction. Pleas modify it to this particular Gh7 and its relevance.

6-   The conclusions are a summary of the results. Can authors elaborate more?

Minor issues:

Throughout the manuscript TtCel7B, Tt should be in italics (as it makes reference to the species)

Page 1, second sentence: “It is may obtain…” , surely authors intended to say: It is obtained or it may be obtained.

Page 2, third paragraph: “It is responsible..” Please say what that It refers to. “This activity..”?

P2, fourth paragraph: where it says “due to its”…it should say “due to their”

Page 7 (2.3.1): activity on xylan or arabinan (the or is missing)

Page 9: 2.4: …systems have a wide range… (instead of has)

Page 10, 3.1: in which assay were Silica plates used?

Page 11, 3.4: please indicate which controls were used in determining activity (enzyme without substrate and substrate without enzyme?)

Author Response

Manuscript ID: catalysts-2249329

RESPONSES TO REVIEWER 1     

First, we would like to thank the reviewer 1 for investing your valuable time in reviewing our manuscript. We list the reviewer' comments by numbering them according to your sequence in the reviewer' questions and your respective answers for better understanding. Our replies are in blue font.

Reviewer #1:

This manuscript reports the recombinant expression in the Aspergillus nidulans platform of a GH7 endoglucanase from the fungus Therthielavioides terrestris and its biochemical characterization. The manuscript is straightforward, and it is very well written.

However, there are some issues that need to be addressed before publishing.

Most relevant issues:

  1. Why was this particular sequence selected? Is it the only GH7 endoglucanase of the fungus? Please include some information in the introduction.

A: Recent studies based on genomic, transcriptomic, and proteomic analyses showed that Thermothielavioides terrestris can secrete an arsenal of CAZymes when cultivated in cellulosic substrates (https://doi.org/10.1186/s13068-021-01975-1 | https://doi.org/10.1038/s41598-019-40213-5). Among the CAZymes, different hydrolases GH7 (endo- and exo-glucanases) were identified and annotated depending on the strain of T. terrestris analyzed (https://doi.org/10.1186/s13068-021-01975-1 | https://doi.org/10.1038/s41598-019-40213-5). In the case of the strain UAMH 3264 utilized in this work, we identified at last four putative endoglucases GH7 not characterized from transcriptomic analysis carried out in our group. Thus, the current work reports the complete characterization of TtCel7B, which offered excellent results to hydrolyse of biomass. To comply with the reviewer’s request and make the text clearer, we added the following sentence in the revised manuscript: Among the CAZymes, different hydrolases GH7 (endo- and exo-glucanases) were identified and annotated depending on the strain of T. terrestris analyzed. (https://doi.org/10.1186/s13068-021-01975-1 | https://doi.org/10.1038/s41598-019-40213-5).

  1. Regarding the expression (section 2.2): was it expressed with its native signal peptide? Or the mature version, with the signal peptide of the vector?

A: Thanks for the note. Since the publication of pEXPYR vector article in 2012 [39], we started to work with some oxidative enzymes that need to have a correct post-translational modification to be in its active mature form. So, we implemented a high throughput method based in Gibson Assembly and it keep the native signal peptide of client protein. To comply with the reviewer’s request and make the text clearer, we added the following sentence in the revised manuscript, page 5: This expression system allows the overexpression along with high-yield secretion and accumulation of the recombinant client protein with its native signal peptide in the culture supernatant [39].

  1. Segato, F.; Damásio, A.R.L.; Gonçalves, T.A.; de Lucas, R.C.; Squina, F.M.; Decker, S.R.; Prade, R.A. High-Yield Secretion of Multiple Client Proteins in Aspergillus. Enzyme Microb. Technol. 2012, 51, 100–106, doi:10.1016/j.enzmictec.2012.04.008
  1. It is important to deposit or include the complete sequence of the recombinant protein. This will help to verify the results shown in Table 1 (mas spec peptides).

A: We comply with the reviewer’s request. The complete sequence of TtCel7B was added in Table 1. Moreover, Table 1 was removed from the original manuscript and added to the supporting file as suggested by reviewer 2#. Please see Supplementary Table S1 in the revised supplementary data file.

  1. Methods 3.3: Why was DNA used? Wasn´t the sequence amplified from RNA? What about the introns? It does not make sense. Please revise The last sentence of page 9 is a repetition of the introduction. Please modify it to this particular Gh7 and its relevance.

A: Thanks for the note. Our protein expression system is a eukaryotic based system (filamentous fungi Aspergillus nidulans), that is different from Escherichia coli. Therefore, the system allows the introduction of the genes coding regions containing introns, since it has the capacity to perform post-transcriptional modifications.  In the case of the gene encoding the T. terrestris GH7 cellulase, it shows a 57 bp single intron near to the 3’ region that does not modify the cell processing. Regarding the second part of query, the last sequence in the page 9 was rewritten in the revised manuscript.

  1. The conclusions are a summary of the results. Can authors elaborate more?

A: Thank you for the valuable suggestion. The conclusion was expanded in the revised manuscript as the reviewer requested.

Minor issues:

  • Throughout the manuscript TtCel7B, Tt should be in italics (as it makes reference to the species)

A: Thank you for outlining this. The correction was made in the revised version of the manuscript.

  • Page 1, second sentence: “It is may obtain…” , surely authors intended to say: It is obtained or it may be obtained.

A: The correction was made in the revised version.

  • Page 2, third paragraph: “It is responsible..” Please say what that It refers to. “This activity..”?

A: We apologize for that. The information was added in the revised manuscript: “Endo-1,4-β-glucanases are responsible for releasing...”

  • P2, fourth paragraph: where it says “due to its”…it should say “due to their”

A: The correction was made in the revised version.

  • Page 7 (2.3.1): activity on xylan or arabinan (the or is missing)

A: We apologize for that. The “or” was added in the revised manuscript (2.3.3).

  • Page 9: 2.4: …systems have a wide range… (instead of has).

A: The change was made as requested.

  • Page 10, 3.1: in which assay were Silica plates used?

A: We apologize for that. We do not perform TLCs (Thin layer chromatographies) in the work. We revised and improved the instrumental details in the Materials and methods section of the revised manuscript.

  • Page 11, 3.4: please indicate which controls were used in determining activity (enzyme without substrate and substrate without enzyme?).

A: Thank you for the valuable suggestion. We revised and improved the description in the Materials and methods section of the revised manuscript, as requested.

Reviewer 2 Report

A request regarding the introduction, it would be helpful for the reader to have a figure representing the biomass activities of the various GH listed

 In Fig. 3 the authors state that the experimental molecular weight of the protein is higher due to glycosylation (69 kDa as opposed to the theoretical 48 kDa); in figure 3B, the authors show a band in lane 1 that appears to be higher than in figure A, but this could be due to various factors, what needs to be clarified by the authors is why, following deglycosylation, the experimental molecular weight is not 48 but probably 60? In my opinion, this experiment in this form does not make much sense. In addition, to more accurately determine the molecular weight of your protein, I suggest using size-exclusion chromatography with appropriate markers.

        table 1 could be moved to the supplementary files

         “Circular dichroism spectroscopy analysis of TtCel7B showed a typical β-class protein profile spectrum, with the minimum negative band at 214 nm and absence of a negative band 222 nm (Figure 4A), suggesting a low content of α-helices, [41,47–49].” To give more strength to this statement, it would be ideal to analyze the CD spectra with tools that define the percentage of alpha or beta. An example is this tool https://onlinelibrary.wiley.com/doi/10.1002/pro.4153

         “2.2 Expression and purification of recombinant TtCel7.1 endoglucanase”. this section is very confusing; although the title talks about expression and purification, a structural and pH resistance characterization has also been done; in my opinion, this section should be reorganized and at least divided into two parts.

         “2.3  Characterization assays”.  I might suggest changing the title to 'biochemical characterization'.

         Figure 5B please specify the substrate

         the conclusions must be expanded

Minor problems:

         TtCel7B, please put in italics Tt

         check within the text that species names are in italics

         “Sequence analysis and molecular 3D modeling of TtCel7.1” is this Cel7B? please clarify

Author Response

Manuscript ID: catalysts-2249329

RESPONSES TO REVIEWER 2     

First, we would like to thank the reviewer 2 for investing your valuable time in reviewing our manuscript. We list the reviewer' comments by numbering them according to your sequence in the reviewer' questions and your respective answers for better understanding. Our replies are in blue font.

Reviewer #2:

  1. A request regarding the introduction, it would be helpful for the reader to have a figure representing the biomass activities of the various GH listed.

A: Thanks for the note. The suggestion to improve the introduction with a figure is very interesting. We understand the raised point; however, the aim of this manuscript introduction was just summarized to the reader the enzymes involved in lignocellulosic biomass degradation. In our perception, if the readers want to go deeper into this subject, they could consult suitable reviews published cited in introduction, as for example, the reference 14*. In the review, we detail the enzymes involved in biomass degradation with nice figures to illustrate the role of these enzymes.

*Freitas, E.N.; Salgado, J.C.; Alnoch, R.C.; Contato, A.G.; Habermann, E.; Michelin, M.; Martínez, C.A.; Polizeli, M. L.T.M. Challenges of Biomass Utilization for Bioenergy in a Climate Change Scenario. Biology 2021, 10, doi:10.3390/biology10121277.

  1. In Fig. 3 the authors state that the experimental molecular weight of the protein is higher due to glycosylation (69 kDa as opposed to the theoretical 48 kDa); in figure 3B, the authors show a band in lane 1 that appears to be higher than in figure A, but this could be due to various factors, what needs to be clarified by the authors is why, following deglycosylation, the experimental molecular weight is not 48 but probably 60? In my opinion, this experiment in this form does not make much sense. In addition, to more accurately determine the molecular weight of your protein, I suggest using size-exclusion chromatography with appropriate markers.

A: Thanks for the note. As described, our results indicated that the MW estimated of 69,5 kDa for TtCel7B corresponds to the glycosylated form. In order to estimate the carbohydrate contents of purified TtCel7B after the purification by size-exclusion chromatography, deglycosylation assays were carried out using the endoglycosidase H from Streptomyces plicatus. This endo-H act only in the N-glycosylation sites. Thus, the MW estimated of 65.2 kDa after deglycosylation assays suggest a high number of O-glycosylation in TtCel7B.  To comply with the reviewer’s request and make the text clearer, we reorganized the section 2.2, added the section 2.3.1. “Estimation of carbohydrate content of TtCel7B”, and addressed the point raised by the reviewer: SDS-PAGE (Figure 3B) shows the profile of native and deglycosylated TtCel7B, with a MW estimated of 65.2 kDa, resulting in a carbohydrate content of approximately 6%. These results corroborate with the in-silico analysis performed by NetGlyc 1.0 Server and N-glycosylation sites predicted from the 3D model of TtCel7B (Figure 2). On the other hand, the MW of deglycosylated TtCel7B still higher than the theoretical (48.7 kDa), which suggest a high number of O-glycosylation in TtCel7B, probably along the flexible peptide linker and CBM of the enzyme, a region rich in Thr and Ser residues (Figure 1), potential O-glycosylation sites.

  1. table 1 could be moved to the supplementary files

A: We comply with the reviewer’s request. Table 1 was removed from the original manuscript and added to the supporting file. Please see Supplementary Table S1 in the revised supplementary data file.

  1. “Circular dichroism spectroscopy analysis of TtCel7B showed a typical β-class protein profile spectrum, with the minimum negative band at 214 nm and absence of a negative band 222 nm (Figure 4A), suggesting a low content of α-helices, [41,47–49].” To give more strength to this statement, it would be ideal to analyze the CD spectra with tools that define the percentage of alpha or beta. An example is this tool https://onlinelibrary.wiley.com/doi/10.1002/pro.4153

A: Thank you for the valuable suggestion. In response to this comment, we addressed the point raised by the reviewer, including the suggested reference (reference 51 in the revised manuscript).

  1. “2.2 Expression and purification of recombinant TtCel7.1 endoglucanase”. this section is very confusing; although the title talks about expression and purification, a structural and pH resistance characterization has also been done; in my opinion, this section should be reorganized and at least divided into two parts.

A: Thank you for the valuable suggestion. The section 2.2 was reorganized and the “Estimation of carbohydrate content of TtCel7B” and “Circular dichroism (CD) and determination of melting temperature (Tm) of TtCel7B” as now part of the section 2.3, leaving the “Expression and purification of recombinant TtCel7B endoglucanase” on the section 2.2.

  1. “2.3 Characterization assays”.  I might suggest changing the title to 'biochemical characterization'.

A: Thank you for the valuable suggestion. The change was made on the revised manuscript as requested. The section 2.3 now reads “Biochemical characterization.”

  1. Figure 5B please specify the substrate

A: We apologize for that. The information now can be found in the revised manuscript on the Figure 5 legend and in the paragraph below.

  1. the conclusions must be expanded

A: Thank you for the valuable suggestion. The conclusion was expanded in the revised manuscript as the reviewer requested.

  1. Minor problems:TtCel7B, please put in italics Tt ; check within the text that species names are in italics; “Sequence analysis and molecular 3D modeling of TtCel7.1” is this Cel7B? please clarify

A: We thank you and agree with the comment. These typographical errors were corrected in the revised manuscript. We also revised the manuscript carefully and corrected any other minor mistakes we have found. The term TtCel7.1 was changed to TtCel7B and corrected in the revised manuscript.

Finally, we revised the manuscript significantly. We tried our best to comply with the reviewers’ questions and suggestions.

Round 2

Reviewer 2 Report

The paper has improved and I thank the authors for considering my suggestions.

Nevertheless, I still think that a summary figure in the introduction would be very effective.